# Fluid-Suppressed Amide Proton Transfer-Weighted Imaging Outperforms Leakage-Corrected Dynamic Susceptibility Contrast Perfusion in Distinguishing Progression from Radionecrosis in Brain Metastases

**DOI:** 10.3390/cancers17071175

**Published:** 2025-03-31

**Authors:** Lucia Nichelli, Stefano Casagranda, Ottavia Dipasquale, Mehdi Bensemain, Christos Papageorgakis, Mauro Zucchelli, Julian Jacob, Charles Valery, Bertrand Mathon, Patrick Liebig, Moritz Zaiss, Stéphane Lehéricy

**Affiliations:** 1Paris Brain Institute (PBI)—Institut du Cerveau (ICM), Sorbonne University, Inserm UMR S 1127, CNRS 7225, F-75013 Paris, France; 2Department of Neuroradiology, La Pitié Salpêtrière Hospital, AP-HP Sorbonne University, F-75013 Paris, France; 3Department of R&D Advanced Applications, Olea Medical, 13600 La Ciotat, France; 4Department of Radiation-Oncology, La Pitié Salpêtrière Hospital, AP-HP Sorbonne University, F-75013 Paris, France; 5Department of Neurosurgery, La Pitié Salpêtrière Hospital, AP-HP Sorbonne University, F-75013 Paris, France; 6Siemens Healthineers AG, 91301 Erlangen, Germany; 7Department of Neuroradiology, Friedrich-Alexander Universität Erlangen-Nürnberg (FAU), 91054 Erlangen, Germany

**Keywords:** APTw imaging, CEST, fluid suppression, radionecrosis, brain metastasis, MTRasym

## Abstract

Assessing a response in an evolving irradiated brain metastasis is difficult and at the same time pivotal for patient care, as it can reflect radionecrosis or tumor progression, two entities with antithetical clinical management. At present, DSC perfusion after leakage correction is the reference MRI technique in this evaluation. Our preliminary study demonstrated that fluid-suppressed APTw better discriminated radionecrosis from tumor progression in comparison to leakage-corrected rCBV imaging. These findings contribute to the expanding body of evidence supporting the added value of APTw imaging in neuro-oncology. However, while there is robust evidence of the potential of APTw in gliomas, studies that include APTw imaging in metastases are scarce, and no study has evaluated a fluid artifact correction approach in APTw maps in therapeutic assessments of brain metastases. This study suggests that fluid-suppressed APTw could serve as a valuable addition to multimodal imaging protocols in post-treatment tumor assessments.

## 1. Introduction

Brain metastases are the most common intracranial tumors in adults. Stereotactic radiosurgery (SRS) is a leading first-line treatment option, as it is a cost-effective, minimally invasive strategy that provides a high level of local control [1]. Following SRS, several complex neuro-inflammatory reactions can occur in the irradiated brain tissue, potentially leading to transient, evolving lesions that can mimic tumor progression. These lesions are commonly grouped under the broad term ‘radionecrosis’. It is estimated that approximately one-third of metastases temporarily increase after SRS. This incidence is likely to rise in the future, following the growing indications for SRS and the synergy between radiotherapy and immunotherapy, which may also increase the rate of treatment-related complications [1].

Distinguishing a radionecrosis complication from a malignant progression in a pre-irradiated evolving lesion represents a persistent, daily, diagnostic difficulty that strongly impacts subsequent patient care. Recurrences require the prompt initiation of novel antitumor treatment strategies (e.g., surgical resection, additional chemotherapy, and repeated radiotherapy) [2,3]. By contrast, symptomatic radiation-induced inflammatory reactions are treated with high-dose steroids and/or bevacizumab [4,5]. However, differentiating between these two entities can be very hard in clinical practice [1].

While conventional magnetic resonance imaging (MRI) features (e.g., morphologic characteristics of T1-weighted and T2-weighted sequences, contrast-enhancing patterns, and lesion quotient [1]) aid clinicians in the evaluation of these lesions, they lack the specificity required to reliably distinguish between radionecrosis and tumor progression. As a result, complementary advanced imaging techniques are warranted. Among these, brain perfusion imaging has an established added value in post-treatment tumor assessments, as it enables clinicians to discriminate between tumoral and non-tumoral contrast enhancing lesions. Three perfusion MRI techniques are available for brain imaging: dynamic susceptibility contrast (DSC), dynamic contrast-enhanced (DCE), and arterial spin labelling (ASL). All of these techniques have shown clinical evidence of their use in the follow-up and post-treatment management of brain tumors, although DCE perfusion has a low cross-site consistency, and ASL is hindered by spatial resolution and by the limited amount of literature on its value in metastases. By contrast, multiple pieces of evidence have shown the reproducibility, repeatability, and clinical value of DSC perfusion, particularly when leakage-correction methods are applied, as recommended by consensus guidelines [6]. To date, relative cerebral brain volume (rCBV), a semi-quantitative vascularity measure derived from DSC perfusion, serves as the cornerstone of MRI-based tumor evaluation. rCBV is a valuable, repeatable, and widely accessible marker that reflects neoplastic neo-angiogenesis, and has a high diagnostic accuracy in post-treatment tumor assessment, especially when combined with diffusion-weighted imaging (DWI), susceptibility-weighted imaging (SWI), and magnetic resonance spectroscopy (MRS) [1].

However, in some clinical contexts, even a comprehensive multimodal MRI protocol with DSC perfusion cannot reliably differentiate between radionecrosis and tumor progression, delaying diagnosis and proper treatment strategies.

Amide proton transfer-weighted (APTw) imaging is a subtype of chemical exchange saturation transfer (CEST) imaging and is currently available on 3T MRI systems [7]. APTw imaging enables in vivo mapping of tissue protein content by detecting variations in the exchange of amide protons, found in mobile peptides and proteins, with water protons. This reliable technique has shown strong potential in neuro-oncology, as elevated protein concentrations in tumors result in increased APTw signal intensity [7].

In pretreatment settings, APTw imaging has already been extensively studied for the baseline characterization of gliomas, demonstrating correlations with tumor grade, molecular biomarkers, histological proliferation indices, and prognosis [8]. Its value as a noninvasive imaging strategy is therefore relevant for lesion characterization, but it cannot replace invasive techniques, as histomolecular diagnostic confirmation remains indispensable for suspected tumor mass lesions.

In post-treatment settings, however, an accurate noninvasive tumor evaluation with APTw imaging could play a pivotal role by aiding clinicians in differentiating between post-therapeutic tissue changes and tumor progression. Tumor hypercellularity is known to increase APTw signal intensity, whereas therapeutic remnants, characterized by lower cellular density, exhibit reduced APTw signal intensity. In gliomas, the high performance of this technique in post-treatment evaluation has been repeatedly and consistently reported, while in brain metastasis, scarcer evidence limits the possibility to draw conclusions about its applicability [9].

Of note, necrotic lesions are frequently seen in irradiated tissues [10], and fluid components alter APTw signals due to a spillover dilution effect [11] This phenomenon leads to APTw signal hyperintensities in fluid compartments, potentially compromising its ability to reliably discriminate between radionecrosis and tumor progression. Recently, fluid-suppression techniques have been shown to correct this artifact [12] and may resolve this limitation.

In this study, we aimed to compare the diagnostic performance of leakage-corrected DSC perfusion with that of fluid-suppressed APTw imaging in differentiating between radionecrosis and tumor progression in evolving, SRS-treated metastases.

## 2. Materials and Methods

### 2.1. Population

We prospectively recruited subjects at a tertiary medical center (Pitié-Salpêtrière University Hospital, Paris, France) based on the following inclusion criteria: (a) age > 18 years, (b) brain metastatic disease, and (c) enlarging lesion following treatment with Gamma Knife SRS. Lesions with a maximum diameter smaller than 3 mm were excluded.

For each consecutively selected subject, we anonymously recorded demographic data (age, sex), primary tumor type, and previous radiotherapy history (type, dose). All participants finished the study. Because this was a pilot study, a formal power calculation was not required.

Subjects were classified into two groups, radionecrosis (RN) or tumor progression (TP), based on the current gold standard for diagnosis, i.e., either by histological examination or with imaging follow-up. In case of the absence of histological examination and absence of conclusive imaging follow-up data, subjects were classified as RN or TP following the reports of the multidisciplinary cancer care team.

This study follows the Strengthening Reporting of Observational Studies in Epidemiology (STROBE) guidelines [13]. The data were made available by the Clinical Data Warehouse of the Public Assistance Hospitals Paris (Assistance Publique Hopitaux Paris, AP-HP, reference number: 20211216174611), in accordance with the Ethical and Scientific Board of the AP-HP. According to AP-HP regulations and the Ethical and Scientific Board of our institution, Institutional Review Board approval and a written consent were waived, as MRIs were acquired as part of routine clinical care.

### 2.2. MRI Acquisition

Patient MRI data were acquired using a 3T MR scanner (MAGNETOM Skyra, Siemens, Erlangen, Germany) with a 64-channel head and neck coil. The APTw protocol (WIP816B, 3:07 min, resolution of 1.7 × 1.7 × 5 mm^3^, 12 slices, acquisition matrix of 128 × 104, repetition time of 4.5 ms, echo time of 2 ms) was performed with a 3D snapshot-gradient echo sequence, with a B1 mean value of 2.22 μT and a duty cycle of 55%, and at 26 equally spaced offsets from −6 ppm to 6 ppm from water frequency, with an additional volume at 300 ppm for APTw data normalization. These were the default protocol settings of the WIP before the publication of the APTw consensus paper [7]. The WASAB1 protocol (WIP816B, 2:03 min, resolution of 1.7 × 1.7 × 5 mm^3^, 12 slices, acquisition matrix of 128 × 104, repetition time of 4.5 ms, echo time of 2 ms) was performed for simultaneous B0 and B1 mapping [14]. DSC perfusion (1:39 min, resolution of 1.8 × 1.8 × 3 mm^3^, 30 slices, acquisition matrix of 120 × 120, repetition time of 1770 ms, echo time of 25 ms) was acquired after a single dose of gadoteric acid contrast agent (0.2 mmol/kg) and a low flip angle (1:39 min, 1.8 × 1.8 × 3 mm^3^, 30 slices), and was followed by 3D FLAIR and 3D T1w spin echo.

### 2.3. MRI Data Post-Processing

Olea Sphere 3.0 software (Olea Medical, La Ciotat, France) was used to process the APTw, WASAB1, and DSC perfusion data. More precisely, it was used to do the following:Compute leakage-corrected relative rCBV maps from DSC perfusion;Compute B0 and B1 maps from the WASAB1 sequence [15];Correct APTw data for noise [16], motion [17], and B0 [7] and B1 [18] artifacts, and then compute fluid-suppressed APTw maps [12];Co-register fluid-suppressed APTw and leakage-corrected rCBV maps with structural sequences;Delineate regions of interest (ROIs) in the lesion and in the contralateral normal appearing white matter (cNAWM).

Voxelwise APTw maps were computed using the magnetization transfer ratio asymmetry (MTRasym) formula at +3.5 ppm relative to the water resonance frequency, based on the principle that amide protons resonate with an offset Δ*ω* of +3.5 ppm downfield from water. The MTRasym calculation, which compares the saturation effects at ±3.5 ppm, is currently the recommended post-processing method for APTw signal intensity quantification (Equation (3) in [7]). Subsequently, APTw maps were corrected for fluid-related artifacts by applying the spillover-based fluid suppression factor to the MTRasym formula, as in Equation (5) in [12].

ROIs were manually drawn on APTw maps co-registered with 3D-FLAIR images by a senior radiologist (L.N.) who was blinded to the clinical information and included the amide signal intensity of the lesion visible on the FLAIR sequence. ROIs were drawn on all slices of lesion visibility on the FLAIR sequence.

### 2.4. Statistical Analyses

Categorical variables were expressed as numbers (%) and compared using χ^2^ tests.

Leakage-corrected rCBV values and fluid-suppressed APTw metrics were considered continuous variables and expressed as mean ± standard deviation (SD).

Data normality was assessed using Shapiro –Wilk tests, after which group comparisons between RN and TP patients were performed using independent samples *t*-tests (Student’s for normally distributed data and Mann–Whitney for non-normal data) on the following normalized metrics:difference in the average fluid-suppressed APTw values calculated for the ROIs, defined as ΔAPTw = APTw_lesion_ − APTw_cNAWM_;ratio of the average leakage-corrected rCBV values calculated for the ROIs, expressed as ΔrCBV = rCBV_lesion_/rCBV_cNAWM_.

The effect size for each comparison was evaluated using Cohen’s d or rank biserial correlation, depending on the *t*-test applied, to quantify the magnitude of group differences and provide additional insight into their significance, regardless of sample size.

The diagnostic performance of the ΔAPTw and ΔrCBV metrics was assessed by calculating the areas under the curve (AUCs) using receiver operating characteristic analysis. For each metric, the accuracy, sensitivity, specificity, positive predictive value, negative predictive value, and Youden index were calculated. 

Statistical significance was defined as *p* < 0.05.

## 3. Results

### 3.1. Characteristics of Subjects

Twenty subjects were prospectively recruited (14 females, 6 males; median age: 60 years; IQR: 54–69; Table 1) from January 2021 to July 2021. Among 21 evolving lesions, 10 (48%) were evaluated as RN and 11 (52%) as TP.

The final diagnosis was established through histological evaluation in 6 out of 21 lesions (29%) and MRI follow-up in 11 out of 21 cases (52%). A histological examination was obtained after total resection of the enhancing lesion component for all the surgically treated patients and was conclusive in all cases. For 4 lesions (9%), the diagnostic uncertainty remained high during MRI follow-up, and the lesions were assessed as RN or TP in accordance with a multidisciplinary oncological meeting.

Table 1 details the clinical characteristics of the population, including primary tumor histology, the administered SRS radiation dose, and the delay between the SRS and the appearance of an evolving lesion. Table 1 also displays subsequent patient care after RN or TP diagnosis for each lesion.

### 3.2. Ability of Fluid-Suppressed APTw and Leakage-Corrected rCBV to Distinguish Between RN and TP

The mean (±SD) values of ΔAPTw and ΔrCBV for the RN and TP groups are shown in Figure 1. Given the non-normality of the samples, nonparametric Mann–Whitney U *t*-tests were used to assess the differences between the RN and TP groups. Fluid-suppressed APTw showed a significant difference between RN and TP (U = 120, *p* < 0.001), while leakage-corrected rCBV did not show significant between-group differences (U = 71, *p* = 0.174). This result is evidence that rCBV is the current benchmark for the distinction between RN and TP. The effect sizes were calculated using rank-biserial correlation, which yielded a large effect for ΔAPTw (r_b_ = 0.983) and a small effect for ΔrCBV (r_b_ = 0.174).

Figure 2 shows an example of APTw and rCBV maps in cases of RN and TP.

### 3.3. Accuracy of Fluid-Suppressed APTw and Leakage-Corrected rCBV Measurements

ROC curves for the two metrics (ΔAPTw and ΔrCBV) are shown in Figure 3. The AUC was 0.991 for ΔAPTw (95% CI = 0.962–1.020) and 0.636 for ΔrCBV (95% CI = 0.352–0.921).

The optimal cutoff points were 0.4 for ΔAPTw and 2.1 for ΔrCBV.

ΔAPTw metrics demonstrated a sensitivity of 100% and a specificity of 90% in distinguishing cerebral lesions, whereas ΔrCBV achieved a sensitivity of 63.6% and a specificity of 36.4%.

To further assess the robustness and generalizability of the model, we performed a leave-one-out cross-validation (LOO CV) and repeated the ROC analysis. The LOO CV method provides a more stringent evaluation by testing the model on each individual subject, which helps mitigate potential overfitting and ensures generalizability. The ROC curve generated from the LOO CV demonstrated an AUC of 0.97 (95% CI = 0.87–1), reflecting strong discriminatory ability. Additionally, a balanced accuracy of 0.90 and an F1 score of 0.88 confirmed that the model performs effectively in classifying both RN and TP, offering a more comprehensive evaluation of the model’s diagnostic performance in a clinical context.

## 4. Discussion

Our study provides encouraging evidence of the role of APTw imaging in brain metastasis follow-up by demonstrating (i) the high diagnostic accuracy of fluid-suppressed APTw imaging in distinguishing between radionecrosis and tumor progression, two entities with antithetical clinical management, and (ii) its increased superiority in this distinction compared to leakage-corrected rCBV, the current imaging benchmark.

Preliminary studies have already suggested the potential of APTw imaging in this setting. Initial results conducted in glioma rat models have shown that an amide-related signal derived from APTw imaging was able to differentiate radiation necrosis from tumor progression [19]. A successive study conducted on glioma rat models before and after radiation observed that APTw provides reliable metrics for response assessments, and that APTw signal intensity decreases earlier than ADC and relative blood flow values derived from ASL [20]. These preclinical findings first suggested the clinical benefits of translating this MRI technique into clinical practice. Subsequent research validated these findings on human subjects with image-guided stereotactic biopsies [21]. This study, which was conducted in patients with a possible progressing diffuse glioma after chemoradiation, found a significant positive correlation between APTw signal intensity and both cellularity and proliferation index, as well as a high accuracy of APTw values for identifying glioma recurrence. Parallel to this study, other studies reported significantly higher APTw intensity values in recurrent gliomas in comparison to treatment-related changes [9], and a superior discriminative power of APTw imaging in post-therapeutic contexts in comparison to perfusion MRI [22], DWI [23], MRS [24], and amino acid PET [25].

In brain metastases, evidence of the reliability of APTw imaging for therapy response assessments is scarce. Two previous studies from the same group reported that the magnetization transfer ratio of amides, the nuclear Overhauser effect, and the relaxation-compensated APT were significant metrics in the distinction between radionecrosis and tumor progression [26,27]. Recently, a monocentric study of 73 patients suggested that CEST and magnetization transfer-derived parameters better discriminated radionecrosis from true progression than rCBV with a 51 min protocol [28]. Our results support these findings but demonstrate that highly robust results can be achieved using fluid-suppressed APTw imaging with a significantly shorter 5 min protocol that includes B0 and B1 mapping.

Brain fluid accumulation is common and multifactorial after radiotherapy. It can result from endothelial injury and consequent edema, but also from coagulative and fibrinoid necrosis, vascular ectasia, and petechiae [10]. Vascular endothelial damage is a major histological finding in irradiated brain lesions, whatever the irradiated tumor lesion type, and is often entangled with demyelination phenomena and neuronal death [29,30,31]. This liquid-rich compartment influences APTw signal intensity, as fluids artificially increase APTw maps computed using standard MTR_asym_ metrics due to spillover dilution effects [11]. Fluid bright artifacts impact the APTw image’s reliability, and probably explain the inconsistent performance of APTw in tumor-mimicking pathologies [32]. Lately, a fluid-suppressed APTw imaging method based on *MTR_Rex_* metrics and spillover correction principles has been proposed, theoretically explained, and validated [12]. Our findings demonstrated that fluid-suppressed APTw can distinguish between radionecrosis and progression with high accuracy, outperforming rCBV in this distinction, and improving previously reported diagnostic performances of APTw imaging without fluid suppression in the same clinical context [9,26,27].

Leakage-corrected rCBV imaging is widely considered the most useful advanced imaging metric in post-therapeutic tumor assessments [1]. This work confirms the utility and current cutoff values of this metric but also highlights the limitations of rCBV in the distinction between radionecrosis and tumor progression in evolving metastases after SRS, as recently reported [28]. Unwanted susceptibility effects are a well-known limitation of DSC perfusion and consequently of rCBV. Susceptibility artifacts are common after radiotherapy, due to radiation-induced microhemorrhages and vascular malformation [10], and may explain the poor performance of rCBV in our study. More generally, metastases can display reduced brain perfusion values before radiotherapy [33], intrinsically limiting the role of perfusion in an early detection of tumor progression and reinforcing the need to complement MRI protocols with a non-perfusion-based sequence. Other advanced, non-perfusion-based techniques exist, such as DWI, SWI, and MRS, but none of them has showed superiority over DSC perfusion after brain radiotherapy [1]. Therefore, the superior diagnostic performance of APT compared to rCBV demonstrated by our work on brain metastases is relevant. 

In addition, while there is robust evidence of the potential of APTw in glioma treatment assessments, multi-parametric studies that include APTw imaging in metastases are scarce [9].

This work is not without limitations. Despite these promising findings, the lack of histopathological diagnoses for all lesions, the small number of cases, and the single center design of this study limit the generalizability of our results. The small sample size reduces the statistical power of our analyses but is mitigated by the robust performance of our model with LOO CV, which demonstrated a high AUC of 0.97, indicating that the model did not overfit and performed consistently across individual subjects. Additionally, the statistical significance of the findings is further supported by the effect sizes, which were large for the fluid-suppressed APTw metric, reinforcing the clinical relevance of the results despite the small sample. Future studies with larger cohorts will be needed to validate these findings and confirm the robustness and reproducibility of the observed associations. Additionally, our APTw sequence had a saturation duty cycle of 55%. Although this duty cycle meets the standard prerequisites for performing APTw imaging (see Equation (4) in [7]), a duty cycle higher than 90% is considered optimal (see Equation (6) in [7]). We did not set the duty cycle to this optimal value because this study was conducted before the publication of APTw consensus guidelines. Large-scale multicenter studies are needed to validate the performance of fluid-suppressed APTw imaging with an optimal [7] duty cycle higher than 90%, should evaluate the impact of possible confounders (e.g., lesion size, location, and primary tumor histology), and should require histological specimen validation. To date, the implementation of APTw imaging in routine clinical practice is hindered by the limited market availability of the dedicated pulse sequence and the post-processing software. Additional evidence of the correlation between APTw metrics and histological findings is also needed, especially in post-therapeutic contexts. However, it should be noted that a binary distinction between tumor progression and radionecrosis is sometimes impossible even at the pathological level, as multiple tangled pathological processes may coexist with tumoral proliferating cells. This multilayered complexity, together with our positive results, and, above all, the paramount clinical consequences of these diagnoses, underscore the need for further advancements in this field.

## 5. Conclusions

Noninvasive in vivo imaging of mobile protein content in an irradiated tissue is feasible and accurate after the fluid suppression of APTw images. In addition, the fluid-suppressed APTw metric outperforms leakage-corrected rCBV in distinguishing between radionecrosis and tumor recurrence in evolving SRS-treated metastases.

These findings support the growing body of evidence of the clinical value of fluid-suppressed APTw imaging, highlighting its potential as a complementary tool in multimodal MRI-based tumor follow-ups. Fluid-suppressed APTw implementation in the clinic may avoid improper or delayed diagnoses and lead to more precise therapeutic decisions.

## Figures and Tables

**Figure 1 cancers-17-01175-f001:**
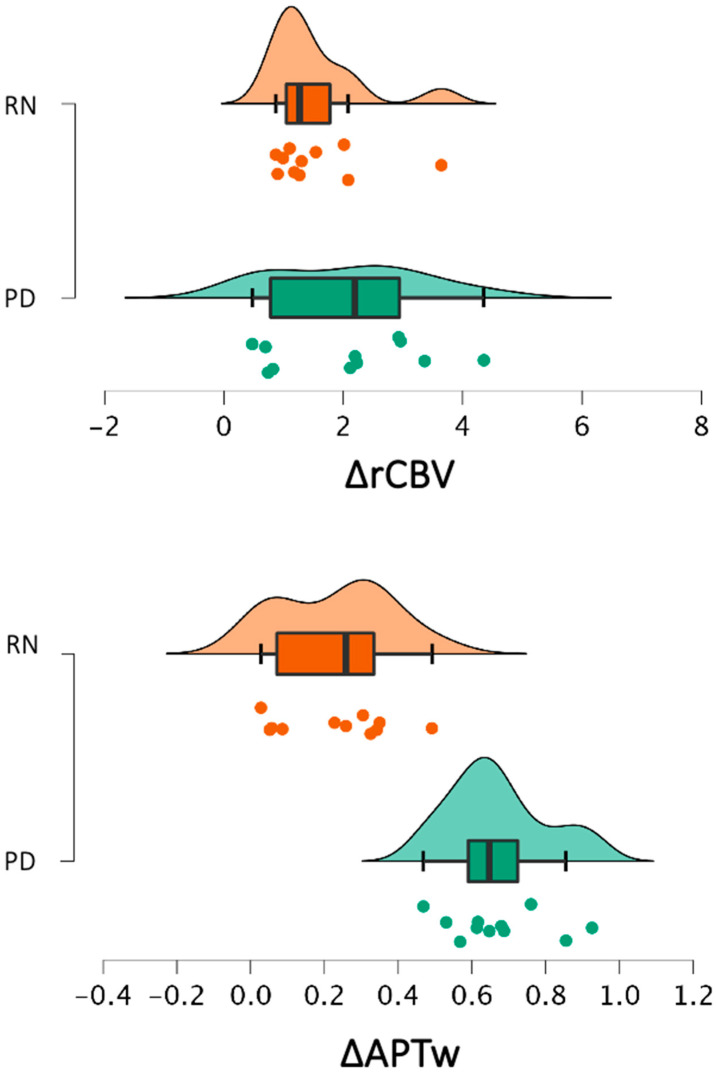
The figure reports raincloud plots to display the distribution of leakage-corrected ΔrCBV and fluid-suppressed ΔAPTw values for the RN and TP groups. Each raincloud plot integrates a density plot (the ‘cloud’), which illustrates the central tendency and variability of the dataset, and scatterplots below display individual data points for each subject. The boxplots within the raincloud plots show the median (central line), the 25th to 75th percentile range (box), and whiskers extending to 1.5 times the interquartile range.

**Figure 2 cancers-17-01175-f002:**
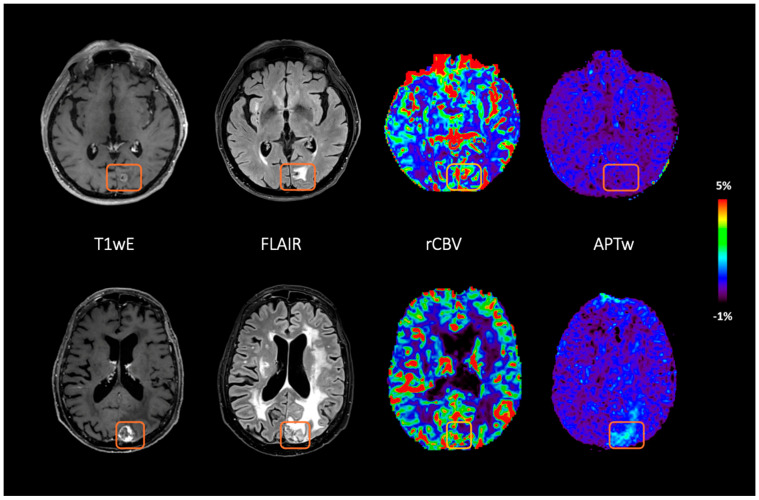
Example of two evolving lesions after brain irradiation (orange box) that corresponded to radionecrosis (upper row, subject S13), confirmed by more than one year of MRI follow-up, and tumor progression (lower row, subject S12), confirmed by histological evaluation. From right to left, axial images of T1 − weighted sequence after contrast injection, axial 3D FLAIR, and leakage-corrected rCBV and fluid-suppressed APTw maps.

**Figure 3 cancers-17-01175-f003:**
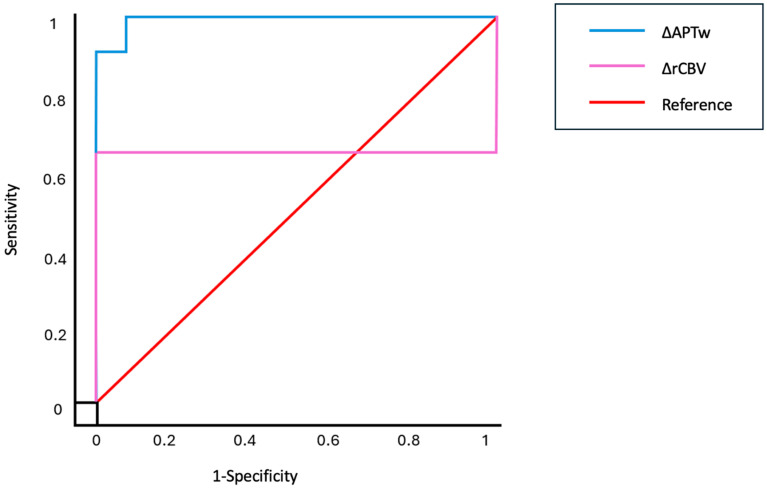
Receiver operating characteristic curves for the prediction of radionecrosis and tumor progression based on fluid-suppressed APTw and leakage-corrected rCBV metrics.

**Table 1 cancers-17-01175-t001:** Characteristics of the enrolled subjects. Abbreviations: F = female; F.S.APT = fluid-suppressed amide proton transfer; Gy = gray; L.C.rCBV = leakage-corrected relative cerebral blood volume; L = lesion; M = male; MDT = multidisciplinary tumor board; N.A. = not available; RN = radionecrosis; SRS = stereotactic radiosurgery; S = subject; TP = tumor progression.

Subject	Sex	Age (Years)	Primary Tumor	SRS (Median Dose/ Number of Fraction)	Previous Radiation	SRS to MRI (Years)	Average F.S.APT	Average L.C.rCBV	Subsequent Patient Care	Diagnosis	Evaluation Method	Follow-Up Time (Years)
S01	M	47	Lung	27 Gy/3	No	0.93	0.34	1.12	Follow-up	RN	Follow-up	2.30
S02	F	44	Germinal tumor	27 Gy/1	No	3.15	0.53	0.74	Re-irradiation	TP	Follow-up	3.22
S03	F	55	Breast	20 Gy/1	No	0.24	0.61	0.48	Re-irradiation	TP	Histology	2.85
S04	F	44	Breast	20 Gy/1	No	3.29	0.47	2.93	Re-irradiation	TP	Follow-up	3.64
S05	M	69	Lung	N.A.	N.A.	0.49	0.09	1.26	Follow-up	RN	MDT	1.00
S06	M	63	Melanoma	N.A.	SRS, 1 fraction	0.51	0.85	2.23	Re-irradiation	TP	Histology	2.23
S07	F	49	Lung	20 Gy/1	No	0.71	0.57	3.36	Re-irradiation	TP	Histology	3.73
S08	F	64	Lung	N.A.	SRS, 2 fractions	2.84	0.93	2.19	Follow-up	TP	Follow-up	1.13
S09	F	59	Breast	27 Gy/3	No	2.82	0.23	0.87	Follow-up	RN	Follow-up	3.49
S10	M	54	Lung	20 Gy/1	No	1.81	0.35	1.30	Follow-up	RN	Follow-up	2.45
S11	F	86	Neuroendocrine	20 Gy/1	No	0.47	0.26	1.54	Follow-up	RN	MDT	0.19
S12	F	72	Lung	20 Gy/2	No	1.48	0.62	0.69	Follow-up	TP	Histology	N.A.
S13	M	80	Melanoma	27 Gy/3	No	1.86	0.06	2.01	Follow-up	RN	Follow-up	1.65
S14	F	68	Lung	20 Gy/1	No	1.29	0.76	2.12	Immunotherapy	TP	Histology	3.38
S15	F	75	Breast	20 Gy/1	No	3.39	0.65	4.36	Chemotherapy	TP	Follow-up	1.83
S16	F	51	Melanoma	24 Gy/1	No	0.47	0.33	1.10	Follow-up	RN	Follow-up	0.19
S17	M	69	Melanoma	N.A.	N.A.	0.23	0.69	2.08	N.A.	RN	MDT	0.18
S18	F	60	Lung	20 Gy/1	WBRT, SRS, 10 fractions	7.15	0.68	0.82	Re-irradiation	TP	Histology	1.43
S19	F	77	Breast	33 Gy/3	No	2.09	0.31	0.90	Anti-angiogenic	RN	Follow-up	1.17
S20_L1	F	54	Breast	20 Gy/1	No	2.10	0.69	2.96	Re-irradiation	TP	MDT	1.11
S20_L2	F	55	Breast	27 Gy/1	No	0.37	0.05	0.99	Follow-up	RN	Follow-up	1.11

## Data Availability

Data supporting this study’s findings can be obtained from the corresponding author upon reasonable request.

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
