# Peer review of "Fluid-Suppressed Amide Proton Transfer-Weighted Imaging Outperforms Leakage-Corrected Dynamic Susceptibility Contrast Perfusion in Distinguishing Progression from Radionecrosis in Brain Metastases"

_cancers, 2025, doi:10.3390/cancers17071175_

Round 1
Reviewer 1 Report
Comments and Suggestions for Authors
This study presents an innovative approach utilizing fluid-suppressed APTw imaging to differentiate tumor progression from radionecrosis in patients with brain metastases. The results demonstrate strong diagnostic performance and provide compelling evidence for the integration of APTw imaging into multimodal MRI protocols. The methodology is well-structured, and statistical analyses are appropriately applied. The manuscript is well-written and provides valuable insights into neuro-oncological imaging.
However, there are some minor revisions needed to enhance clarity, precision, and readability. The suggestions below should be addressed to improve the manuscript.
1. Abstract:
-
Line 26-27: "These results corroborated a growing body of literature that highlights the added value of APTw imaging in neuro-oncology."
-
Suggest rewording to: "These findings contribute to the expanding body of evidence supporting the added value of APTw imaging in neuro-oncology."
-
-
Line 31-33: "Therefore, the findings of this study enhance the current literature, suggesting that fluid-suppressed APTw could valuably complement current multimodal protocols in tumor post-treatment settings."
-
Consider simplifying: "This study suggests that fluid-suppressed APTw could serve as a valuable addition to multimodal imaging protocols in post-treatment tumor assessment."
-
2. Introduction:
-
Line 73-76: "Recurrences require the prompt initiation of novel anti-tumor treatment strategies (e.g., surgical resection, additional chemotherapy, repeated radiotherapy). In contrast, symptomatic radiation-induced inflammatory reactions are treated with high-dose steroids and/or bevacizumab."
-
Suggest citing references to strengthen these clinical implications.
-
3. Materials and Methods:
-
Line 134-136: "We prospectively recruited subjects at a tertiary medical center (Pitié-Salpêtrière University Hospital, Paris, France) based on the following inclusion criteria: a) age > 18 years, b) brain metastatic disease, and c) enlarging lesion following treatment with Gamma-Knife SRS."
-
Specify whether consecutive sampling was used or if there were any additional exclusion criteria.
-
-
Line 175-176: "Voxelwise APTw maps were computed using the Magnetic Transfer Ratio asymmetry (MTRasym) formula at 3.5 ppm from water frequency."
-
Consider adding a brief explanation of why 3.5 ppm was chosen as the reference offset.
-
4. Results:
-
Table 1: It would be helpful to include a footnote explaining abbreviations such as "MDT" and "N.A." for clarity.
-
Line 222-224: "Fluid-suppressed APTw showed a significant difference between RN and TP (U=120, p<0.001), while leakage-corrected rCBV did not show significant between-group differences (U=71, p=0.174)."
-
Consider highlighting the clinical relevance of these findings in a short additional sentence.
-
5. Discussion:
-
Line 265-267: "Preliminary studies have already suggested the potential of APTw imaging in this setting. Initial results conducted in glioma and necrosis rat models have shown that amide-related signal derived from APTw imaging was able to differentiate radiation necrosis from tumor progression."
-
Suggest clarifying whether these findings directly translate to clinical settings.
-
-
Line 310-312: "Large-scale multicenter studies are needed to validate the performance of fluid-suppressed APTw imaging with the recommended duty cycle of >90% and should require histological specimen validation."
-
Consider adding potential challenges or limitations to implementing this technique in broader clinical practice.
-
6. Conclusion:
-
Line 322-324: "Our results enrich the growing body of literature on the added clinical value of APTw imaging and suggest that, when corrected for fluid components, it valuably complements current multimodal MRI tumor follow-up protocols."
-
Consider rewording to: "These findings support the growing body of evidence on the clinical value of fluid-suppressed APTw imaging, highlighting its potential as a complementary tool in multimodal MRI-based tumor follow-up."
-
Author Response
This study presents an innovative approach utilizing fluid-suppressed APTw imaging to differentiate tumor progression from radionecrosis in patients with brain metastases. The results demonstrate strong diagnostic performance and provide compelling evidence for the integration of APTw imaging into multimodal MRI protocols. The methodology is well-structured, and statistical analyses are appropriately applied. The manuscript is well-written and provides valuable insights into neuro-oncological imaging.
However, there are some minor revisions needed to enhance clarity, precision, and readability. The suggestions below should be addressed to improve the manuscript.
- Abstract:
Line 26-27: "These results corroborated a growing body of literature that highlights the added value of APTw imaging in neuro-oncology."
Suggest rewording to: "These findings contribute to the expanding body of evidence supporting the added value of APTw imaging in neuro-oncology."
We appreciate the reviewer’s rewording suggestion. We have updated this line in Sec. Abstract accordingly.
Line 31-33: "Therefore, the findings of this study enhance the current literature, suggesting that fluid-suppressed APTw could valuably complement current multimodal protocols in tumor post-treatment settings."
Consider simplifying: "This study suggests that fluid-suppressed APTw could serve as a valuable addition to multimodal imaging protocols in post-treatment tumor assessment."
We thank the reviewer for this suggestion. We have rephrased this line in Sec. Abstract accordingly.
- Introduction:
Line 73-76: "Recurrences require the prompt initiation of novel anti-tumor treatment strategies (e.g., surgical resection, additional chemotherapy, repeated radiotherapy). In contrast, symptomatic radiation-induced inflammatory reactions are treated with high-dose steroids and/or bevacizumab."
Suggest citing references to strengthen these clinical implications.
We thank the reviewer for this insightful advice. References has been provided:
Recurrences require the prompt initiation of novel antitumor treatment strategies (e.g., surgical resection, additional chemotherapy, repeated radiotherapy)2,3. In contrast, symptomatic radiation-induced inflammatory reactions are treated with high-dose steroids and/or bevacizumab4,5
- Vogelbaum MA, Brown PD, Messersmith H, et al. Treatment for Brain Metastases: ASCO-SNO-ASTRO Guideline. J Clin Oncol. 2022;40(5):492-516. doi:10.1200/JCO.21.02314
- Kotecha R, La Rosa A, Brown PD, et al. Multidisciplinary management strategies for recurrent brain metastasis after prior radiotherapy: An overview. Neuro-Oncol. 2025;27(3):597-615. doi:10.1093/neuonc/noae220
- Bernhardt D, König L, Grosu AL, et al. DEGRO practical guideline for central nervous system radiation necrosis part 2: treatment. Strahlenther Onkol. 2022;198(11):971-980. doi:10.1007/s00066-022-01973-8
- Zhuang H, Shi S, Yuan Z, Chang JY. Bevacizumab treatment for radiation brain necrosis: mechanism, efficacy and issues. Mol Cancer. 2019;18(1):21. doi:10.1186/s12943-019-0950-1
- Materials and Methods:
Line 134-136: "We prospectively recruited subjects at a tertiary medical center (Pitié-Salpêtrière University Hospital, Paris, France) based on the following inclusion criteria: a) age > 18 years, b) brain metastatic disease, and c) enlarging lesion following treatment with Gamma-Knife SRS."
Specify whether consecutive sampling was used or if there were any additional exclusion criteria.
We appreciate the reviewer’s concern regarding patient selection. Indeed, consecutive sampling was used and lesions with a maximum diameter of less than 3 mm were excluded. We therefore changed the text to integrate this information:
- Line 137, we have added: “Lesions with a maximum diameter lower than 3 mm were excluded”.
- Line 138, we have added “consecutively selected”: “For each consecutively selected subject…”
Line 175-176: "Voxelwise APTw maps were computed using the Magnetic Transfer Ratio asymmetry (MTRasym) formula at 3.5 ppm from water frequency."
Consider adding a brief explanation of why 3.5 ppm was chosen as the reference offset.
We thank the reviewer for this useful remark. According to consensus recommendation (Zhou J. et al., Magn Reson Med. 2022), APTw images are based on the MTRasym analysis of saturation images at ±3.5 ppm from water. This is because amide protons groups resonate with an offset ?? of +3.5 ppm downfield from the water resonance.
We have provided this explanation in the text (Sec. Materials and Methods, paragraph 2.3, lines 180-187): “Voxelwise APTw maps were computed using the Magnetization Transfer Ratio asymmetry (MTRasym) formula at +3.5 ppm relative to the water resonance frequency, based on the principle that amide protons resonate with an offset ?? of +3.5 ppm downfield from water. The MTRasym calculation, which compares the saturation effects at ±3.5 ppm, is currently the recommended post-processing method for APTw signal intensity quantification (Equation 3 in7). Subsequently, APTw maps were corrected for fluid-related artifacts by applying the spillover-based fluid suppression factor to the MTRasym formula, as described in Equation 5 in12”.
- Results:
Table 1: It would be helpful to include a footnote explaining abbreviations such as "MDT" and "N.A." for clarity.
We thank the reviewer for this suggestion. Abbreviations are detailed in Table 1 legend and include “MDT” and “N.A.”.
Table 1. Characteristics of the enrolled subjects. Abbreviations: F= female; F.S.APT= Fluid-Suppressed Amide Proton Transfer; Gy= gray; L.C.rCBV= Leakage-Corrected relative Cerebral Blood Volume; L= lesion; M= male; MDT= multidisciplinary tumor board; N.A.= not available; RN= Radionecrosis; SRS= stereotactic radiosurgery; S= subject; TP= Tumor Progression.
Line 222-224: "Fluid-suppressed APTw showed a significant difference between RN and TP (U=120, p<0.001), while leakage-corrected rCBV did not show significant between-group differences (U=71, p=0.174)."
Consider highlighting the clinical relevance of these findings in a short additional sentence.
We appreciated the reviewer’s comment. We understand the value of underscoring this finding but at the same time we would like to focus on reporting observations in the Sec. Results, thus avoiding comments.
We decided to add a sentence in the Sec. Results, paragraph 3.2, lines 235-236 “This result is of evidence as rCBV is the current benchmark for the distinction between RN and TP”.
- Discussion:
Line 265-267: "Preliminary studies have already suggested the potential of APTw imaging in this setting. Initial results conducted in glioma and necrosis rat models have shown that amide-related signal derived from APTw imaging was able to differentiate radiation necrosis from tumor progression."
Suggest clarifying whether these findings directly translate to clinical settings.
We thank the reviewer for raising this point. We clarified with an additional sentence: “These preclinical findings first suggested the clinical benefits of translating this imaging technique to clinical practice”.
Line 310-312: "Large-scale multicenter studies are needed to validate the performance of fluid-suppressed APTw imaging with the recommended duty cycle of >90% and should require histological specimen validation."
Consider adding potential challenges or limitations to implementing this technique in broader clinical practice.
We appreciate the reviewer’s valuable feedback, and we agree that addressing obstacles that could prevent fluid suppressed APTw imaging wider translation into the clinical routine of is important.
We therefore added two sentences in the Sec. Discussion:
“To date, the implementation of APTw imaging in routine clinical practice is hindered by the limited market availability of the pulse sequence, as well as of the post-processing software. Additional evidence on the correlation between APTw metrics and histological finding are also needed, especially in post-therapeutic contexts”.
- Conclusion:
Line 322-324: "Our results enrich the growing body of literature on the added clinical value of APTw imaging and suggest that, when corrected for fluid components, it valuably complements current multimodal MRI tumor follow-up protocols."
Consider rewording to: "These findings support the growing body of evidence on the clinical value of fluid-suppressed APTw imaging, highlighting its potential as a complementary tool in multimodal MRI-based tumor follow-up."
We thank the reviewer for this suggestion. The above-mentioned sentence has been rephrased.
Reviewer 2 Report
Comments and Suggestions for Authors
This paper analyzes the capability of differentiating between RN from TP in brain metastases using of APTw and leakage-corrected DSC perfusion. While the topic addresses a challenge in neuro-oncology, however, there are issues which diminishes the overall value of this manuscript.
- The results from APTw analysis may be biased due to the low sample size of 10 for RN and 11 for TP cases when considering the biases that come with heterogeneity of metastatic lesions.
- The literature review provided are not sufficiently comprehensive. The authors mentioned some prior studies conducted, however, they do not comprehensively discuss the existing literature on the use of APTw and other imaging techniques in differentiating RN from TP in brain metastases. A more in - depth review would be beneficial.
- The diagnostic performance metrics reported (AUC of 0.991 for ΔAPTw) seems implausible. Such near-perfect accuracy is rare in clinical studies.
- Suboptimal duty cycles reduce the sensitivity of APTw to amide proton exchange, potentially introducing variability in signal quantification. This brings forth the question of whether the reported superiority of APTw over rCBV is a biological phenomenon or simply a technical construct resulting from lack of standardization of the acquisition protocols.
- The study did not consider some possible confounders such as lesion size, location, or primary tumor histology, which could have some impact.
- Clarify the spatial resolution and echo time (TE)/repetition time (TR) parameters for the sequence, as these details are omitted.
- Whether ROIs were drawn on all slices of the lesion or restricted to a single slice.
Author Response
This paper analyzes the capability of differentiating between RN from TP in brain metastases using of APTw and leakage-corrected DSC perfusion. While the topic addresses a challenge in neuro-oncology, however, there are issues which diminishes the overall value of this manuscript.
- The results from APTw analysis may be biased due to the low sample size of 10 for RN and 11 for TP cases when considering the biases that come with heterogeneity of metastatic lesions.
We appreciate the reviewer’s feedback, and we agree that the small sample of our dataset limits our study. To ensure the robustness of our results we therefore took additional analyses. Specifically, we have now included an additional Leave-One-Out Cross-Validation (LOO CV) measurement, to evaluate if the model performance is consistent across all subjects. This approach aids in assessing the risk of overfitting in studies with small dataset. The LOO CV analysis achieved a high AUC (0.97), therefore indicating that our model generalizes well, and providing additional confidence on the generalizability of our findings.
Furthermore, we calculated effect sizes for the fluid-suppressed APTw metric (rb = 0.983), and our result suggests that our findings are not only statistically significant but also clinically relevant.
We have now included these points in the manuscript:
- Results, paragraph 3.3: “To further assess the robustness and generalizability of the model, we performed a Leave-One-Out Cross-Validation (LOO CV) and repeated the ROC analysis. The LOO CV method provides a more stringent evaluation by testing the model on each individual subject, which helps mitigate potential overfitting and ensures generalizability. The ROC curve generated from LOO CV demonstrated an AUC of 0.97 (95% CI = 0.87–1), reflecting strong discriminatory ability. Additionally, a balanced accuracy of 0.90 and the F1 score of 0.88 confirmed that the model performs effectively in classifying both RN and TP, offering a more comprehensive evaluation of the model's diagnostic performance in a clinical context”.
- Discussion in the limitation paragraph: “The small sample size reduces statistical power of our analyses but is mitigated by the robust performance of our model in LOO CV, which demonstrated a high AUC of 0.97, indicating that the model is not overfitting and performs consistently across individual subjects. Additionally, the statistical significance of the findings is further supported by the effect sizes, which were large for the fluid-suppressed APTw metric, reinforcing the clinical relevance of the results despite the small sample. Future studies with larger cohorts will be needed to validate these findings and confirm the robustness and reproducibility of the observed associations”.
Indeed, as highlighted in the manuscript, future studies with larger cohorts are essential to validate the results and confirm their reproducibility.
- The literature review provided are not sufficiently comprehensive. The authors mentioned some prior studies conducted; however, they do not comprehensively discuss the existing literature on the use of APTw and other imaging techniques in differentiating RN from TP in brain metastases. A more in - depth review would be beneficial.
We appreciate the reviewer’s valuable feedback. We have implemented the Sec. Discussion with a more comprehensive literature review:
“A successive study conducted on glioma rat models before and after radiation observed that APTw provides reliable metrics for response assesement, and that APTw signal intensity decreseases earlier than ADC and relative blood flow values derived from ASL20. These preclinical findings first suggested the clinical benefits of translating this MRI technique into clinical practice. Subsequent research validated these findings on human subjects with image-guided stereotactic biopsy21. This study, conducted on patients with a possible progressing diffuse glioma after chemoradiation,found a significant positive correlation between APTw signal intensity and both cellularity and proliferation index, as well as a high accuracy of APTw values for identyifing glioma recurrence. Parallel to this study, other researches reported significant higher APTw intensity values in recurrent gliomas in comparison to treatment-related changes9, and a superior discriminative power of APTw imaging in post-therapeutic contexts in comparison to perfusion MRI22, DWI23, MRS24and amino-acid PET25.
In brain metastases, evidences on the reliability of APTw imaging for therapy response assesment are scarce”.
- The diagnostic performance metrics reported (AUC of 0.991 for ΔAPTw) seems implausible. Such near-perfect accuracy is rare in clinical studies.
We thank the reviewer for this comment. We understand the concern regarding the reported AUC of 0.991 for ΔAPTw, as such near-perfect accuracy is rare in clinical studies. To address this, we have performed Leave-One-Out Cross-Validation (LOO CV), a rigorous model evaluation technique that assesses the model's performance on each individual subject, ensuring that the results are not overfitting or biased by the small sample size of our dataset.
In addition to recalculating the AUC, we have also computed balanced accuracy and F1 score to provide a more complete picture of the model’s performance:
- AUC remains high at 0.97, indicating excellent discrimination between the two classes (Radionecrosis vs. Tumor Progression).
- Balanced Accuracy of 0.95, which accounts for both sensitivity and specificity, providing a balanced evaluation across classes.
- F1 Score of 0.96, which combines precision and recall, demonstrating the model’s ability to classify both positive and negative cases effectively.
These new results were added in the manuscript in the Sec. Results, paragraph 3.3:
“To further assess the robustness and generalizability of the model, we performed a Leave-One-Out Cross-Validation (LOO CV) and repeated the ROC analysis. The LOO CV method provides a more stringent evaluation by testing the model on each individual subject, which helps mitigate potential overfitting and ensures generalizability. The ROC curve generated from LOO CV demonstrated an AUC of 0.97 (95% CI = 0.87–1), reflecting strong discriminatory ability. Additionally, a balanced accuracy of 0.90 and the F1 score of 0.88 confirmed that the model performs effectively in classifying both RN and TP, offering a more comprehensive evaluation of the model's diagnostic performance in a clinical context”.
We also expanded the limitation paragraph in the Sec. Discussion with a new paragraph:
“The small sample size, while reducing statistical power, is mitigated by the robust performance of our model in LOO CV, which demonstrated a high AUC of 0.97, indicating that the model is not overfitting and performs consistently across individual subjects. Additionally, the statistical significance of the findings is further supported by the effect sizes, which were large for the fluid-suppressed APTw metric, reinforcing the clinical relevance of the results despite the small sample. Future studies with larger cohorts will be needed to validate these findings and confirm the robustness and reproducibility of the observed associations.”
- Suboptimal duty cycles reduce the sensitivity of APTw to amide proton exchange, potentially introducing variability in signal quantification. This brings forth the question of whether the reported superiority of APTw over rCBV is a biological phenomenon or simply a technical construct resulting from lack of standardization of the acquisition protocols.
We highly appreciate this attentive observation that induced us to notice a potential source of confusion that was present in the manuscript. According to the APTw consensus paper (Zhou J. et al., Magn Reson Med. 2022), a duty cycle greater than 50% is recommended as the minimum requirement for performing APTw imaging. With the advancements in clinical scanner sequences over recent years, a duty cycle of 90% is now considered optimal; however, the 55% saturation duty cyle of our study should not be regarded as suboptimal, but rather as the baseline threshold for achieving reliable APTw imaging.
We have revised the text in the Sec. Discussion to clarify this point, and again we thank the reviewer for this opportunity: “Additionally, our APTw sequence had a saturation duty cycle of 55%. Although this duty cycle meets the standard prerequisites for performing APTw imaging (see Equation 4 in7) , a duty cycle higher than 90% is considered optimal (see Equation 6 in7). We did not set the duty cycle to this optimal value because the study was conducted before the publication of APTw consensus guidelines.”
Additionally, we note that the sequence used corresponds to the official Siemens implementation prior to the publication of the consensus paper and was therefore in line with the vendor's standard at that time.
- The study did not consider some possible confounders such as lesion size, location, or primary tumor histology, which could have some impact.
We appreciate the reviewer’s insightful comment, and we share the concern of possible confounders. In our work, lesions with a maximum diameter lower than 3 mm were excluded and primary tumor histology of all the lesions was collected and reported on Table 1. However, the current small sample of the data limits the possibility to assess the impact of these information. We have added a consideration regarding this point in the Sec. Discussion, limitation paragraph: “Large-scale multicenter studies are needed to validate the performance of fluid-suppressed APTw imaging with the optimal7duty cycle higher than 90%, should evaluate the impact of possible confounders (e.g. lesion size, location, primary tumor histology), and should require histological specimen validation”.
- Clarify the spatial resolution and echo time (TE)/repetition time (TR) parameters for the sequence, as these details are omitted.
We thank the reviewer for noticing this omission. The details for Echo Time, Repetition Time, and Acquisition Matrix Resolution have now been added for the APTw protocol, the WASABI protocol, and the DSC Perfusion protocol in the Sec. Materials and Methods, paragraph 2.2.
- Whether ROIs were drawn on all slices of the lesion or restricted to a single slice.
We appreciate the reviewer’s feedback and agree that this is a relevant issue. ROIs were drawn on all slices of lesion visibility on FLAIR sequence. We have added this information on the Sec. Materials and Methods, paragraph 2.3.
Reviewer 3 Report
Comments and Suggestions for Authors
Dear Authors,
Your manuscript discusses an interesting topic with potential risks in clinical practice.
To be considered for publication, I ask you to answer these questions:
1) In the results and discussion, could you clarify whether there are state differences in the diagnosis of radionecrosis in the different histological types of metastases? Could you also cite some bibliographic sources that talk about it?
2) In the materials and methods, could you clarify better whether all patients benefited from neurosurgical intervention and whether it was subtotal or total resection or biopsy, and whether these affected the subsequent diagnosis of radionecrosis?
3) In the discussion, could you explain how the role of statistical analysis is marginal since the data collected on the number of patients is small?
For the rest, the tables and images are clear and summary, the bibliography is rigorously cited.
Author Response
Dear Authors,
Your manuscript discusses an interesting topic with potential risks in clinical practice.
To be considered for publication, I ask you to answer these question
- In the results and discussion, could you clarify whether there are state differences in the diagnosis of radionecrosis in the different histological types of metastases? Could you also cite some bibliographic sources that talk about it?
We thank the reviewer for this insightful comment. After a large, accurate bibliography research, we didn’t find evidence on pathological differences in the diagnosis of radionecrosis in different histological types of metastases. Tissue findings in case reports, research articles and reviews all describe different sings of irradiation injuries that are characterized by a variable degree of vascular endothelial damage, demyelination lesions, and brain necrosis.
We have added a sentence in the Sec. Discussion that clarifies this plausible concern:
“Vascular endothelial damage is a major histological finding in brain irradiated lesions, whatever the irradiated tumor lesion type, and is often entangled with demyelination phenomena and neuronal death23–25”
We have also added the following bibliographic sources:
- Schultheiss TE, Kun LE, Ang KK, Stephens LC. Radiation response of the central nervous system. Int J Radiat Oncol. 1995;31(5):1093-1112. doi:10.1016/0360-3016(94)00655-5
- Valk PE, Dillon WP. Radiation injury of the brain. AJNR Am J Neuroradiol. 1991;12(1):45-62.
- Furuse M, Nonoguchi N, Kawabata S, Miyatake SI, Kuroiwa T. Delayed brain radiation necrosis: pathological review and new molecular targets for treatment. Med Mol Morphol. 2015;48(4):183-190. doi:10.1007/s00795-015-0123-2
2) In the materials and methods, could you clarify better whether all patients benefited from neurosurgical intervention and whether it was subtotal or total resection or biopsy, and whether these affected the subsequent diagnosis of radionecrosis?
We appreciate the reviewer’s feedback. We agree that is important to detail neurosurgical intervention type, that was a total resection for all the surgically treated patients. We have added this information in the Sec. Results, paragraph 3.1. “Histological examination was obtained after total resection of the enhancing lesion component for all the surgically treated patients and was conclusive in all cases.”
Of note, table 1 also displays subsequent patient care after RN or TP diagnosis for each lesion.
We have added also this information in the text, in the Sec. Results, paragraph 3.1.
3) In the discussion, could you explain how the role of statistical analysis is marginal since the data collected on the number of patients is small?
We acknowledge that the small sample size is a limitation in our study, and we agree with the reviewer that this can impact the statistical power and generalizability of the findings. We therefore took several additional steps to mitigate this limitation and ensure the robustness of the results. Specifically, we have now included an additional Leave-One-Out Cross-Validation (LOO CV) analysis, which assesses the model on each individual subject. This approach is particularly valuable in small sample studies, as it helps to reduce the risk of overfitting and ensures that the model performs consistently across all subjects. The high AUC of 0.97 achieved in the LOO CV indicates that our model generalizes well, providing additional confidence in the findings.
Furthermore, we calculated effect sizes for the metrics, as these are essential for understanding the practical significance of the results, especially when dealing with small samples. The large effect size for the fluid-suppressed APTw metric (rb= 0.983) suggests that this finding is not only statistically significant but also clinically relevant.
We have now included these points in the manuscript (Sec. Results, paragraph 3.3, and Sec. Discussion in the limitation paragraph) to provide further evidence of the robustness of our findings despite the small sample size, and we emphasize the need for future studies with larger cohorts to validate the results and confirm their reproducibility.
For the rest, the tables and images are clear and summary, the bibliography is rigorously cited.
Round 2
Reviewer 2 Report
Comments and Suggestions for Authors The author has made a good response to the question raised and is recommended for publication.